# Evaluation of anti-ulcer activity of hydromethanol crude extract and solvent fractions of *Vicia faba* (Fabaceae) seeds in mice

Demis Getachew[ID]*, Alemante Tafese Beyna[ID], Mohammedbrhan Abdelwuhab, Assefa Belay Asrie

Department of Pharmacology, School of Pharmacy, College of Medicine and Health Sciences, University of Gondar, Gondar, Ethiopia

* demisgetachew1@gmail.com

## Abstract

Peptic ulcer disease affects 10% of the global population, and current treatments have limitations. Vicia faba seeds are traditionally used in Ethiopia for gastric ulcers and gastritis, but scientific evidence is lacking. Seeds were extracted with 80% hydro-methanol to evaluate its antiulcer effect, and part of the extract was fractionated into n-hexane, ethyl acetate, and aqueous solvents. The antiulcer activity of the crude extract and solvent fractions was tested using pylorus ligation and ethanol-induced ulcer models. Repeated dose studies were also performed on both models. In a pylorus ligation-induced ulcer model, single-dose studies showed that 200 mg/kg significantly reduced ulcer number, score ($P < 0.05$), and index ($P < 0.01$), providing 39.83% protection. The 400 mg/kg, significantly reduced ulcer number, and score ($P < 0.01$), providing 55.80% protection. In a repeated-dose study, 100 mg/kg significantly reduced ulcer number and score ($P < 0.05$). The 200 mg/kg dose showed stronger effects ($P < 0.01$, for the above parameters, respectively). The highest dose (400 mg/kg) significantly reduced ulcer parameters ($P < 0.001$) and also lowered gastric juice volume ($P < 0.01$), acidity ($P < 0.05$), and raised pH ($P < 0.01$), providing 59.17% protection. In the ethanol-induced ulcer model, single-dose crude extract provided an ulcer index protection of 23.39%, 34.90%, and 54.04% at 100, 200, and 400 mg/kg, respectively. Repeated administration of the crude extract increased ulcer protection to 34.94%, 50.91%, and 65.72% at the same respective doses. All doses of the aqueous and the ethyl acetate fractions also showed antiulcer activity. This study confirmed that *Vicia faba* seeds crude extract and solvent fractions have significant antiulcer activity, providing scientific evidence supporting the traditional use of *V. faba* seeds as a remedy for gastric ulcers in Ethiopia.

**Data availability statement:** All data used to conduct this study are provided within the manuscript.

**Funding:** Our research is funded by the University of Gondar, Ethiopia. The funder, the University of Gondar, had no role in study design, data collection and analysis, the decision to publish, or the preparation of the manuscript.

**Competing interests:** The authors declare no competing interests.

**Abbreviations:** COX-1, Cyclooxygenase-1; COX-2, Cyclooxygenase-2; DW, Distilled Water; HCl, Hydrochloric acid; H.pylori, Helicobacter pylori; H+/K+ATPase, Hydrogen/ potassium adenosine triphosphatase; NaOH, Sodium hydroxide; NSAIDs, Nonsteroidal anti-inflammatory drugs; PPI, Proton pump inhibitor; PUD, Peptic ulcer disease; SPSS, Statistical Package for the Social Sciences; OECD, Organization for Economic Co-operation and Development; V.faba, Vicia faba.

## Introduction

The word 'peptic' is derived from the Greek term peptikos, which means to digest [1]. Peptic ulcer disease (PUD) is a prevalent digestive system disorder characterized by damage to the digestive tract resulting in a mucosal break exceeding 3–5mm, with visible depth extending into the submucosa [2–4]. The persistence of a peptic ulcer arises from an imbalance between the protective elements of the stomach lining (including mucus and bicarbonate secretion, sufficient blood circulation, prostaglandin E2, nitric oxide, sulfhydryl compounds, and antioxidant enzymes) and aggressive factors (acid and pepsin secretions) [5–7]. The primary factors leading to this imbalance are predominantly attributed to *Helicobacter pylori (H. pylori)* infection and the taking of non-steroidal anti-inflammatory drugs (NSAIDs) [8]. The third important element is stress, which can cause ulcers through various mechanisms, such as increased histamine release, which can raise acid production, and decreased stomach mucus volume and consistency [9].

The current management of PUD consists of proton pump inhibitors (PPIs), histamine H2 receptor antagonists, antacids, the muscarinic (M1) receptor antagonist pirenzepine, antimicrobial agents, and PGE2, along with its analog misoprostol [10,11]. However, these medications have several limitations. They are expensive and carry risks of treatment failure and relapse, drug interactions, and various side effects, including impotence, gynecomastia, galactorrhea, and gastrointestinal infections [10]. All these factors enforce the need to develop new drugs for treating ulcers and to explore unique compounds sourced from the natural drug reservoir, particularly herbal resources.

Traditionally, several prospective medicinal plants for peptic ulcer treatment have been studied and reported to have several advantages: they tend to be safer, more affordable, readily available, and culturally accepted in comparison to modern drugs [12,13]. Ethiopia boasts a wealth of medicinal plants with significant potential for preventing, treating, or managing PUD. Among these, *Vicia faba L. (V. faba),* a member of the Fabaceae family, is an annual legume [14] cultivated in various eco-geographical regions in Ethiopia, including the Arsi and Bale highlands, the Central Highlands (South-West, West, and North Showa), Tigray, North and South Wollo, North and South Gondar, East and West parts of Gojam, Wollega, Guji highlands, Hadia, Sidama, and Gamogofa [15]. It is commonly referred to as a fava bean, broad bean, or horse bean in English [16]. In Amharic and Sidama, it is known as 'Bakella' [17]. Various ethnobotanical studies showed that *V. faba* leaves, roots, sprouts, pods, and beans have been utilized in traditional medicine, either as infusions or dietary components, for the natural management of a range of chronic conditions, encompassing depression [18], Parkinson's disease [19], allergies, diarrhea [20], and stomach ulcers [21]. The plant is recognized for its application in diabetes treatment, given its antioxidant content with free radical scavenging activity, aiding in the rejuvenation of pancreatic beta-cells [22]. Additionally, the plant's fruit has exhibited anticancer properties in individuals with colon cancer [23].

*In vitro*, antimicrobial assays showed that *V. faba* pod extracts inhibited the growth of bacteria such as *B. subtilis, Staph Aureus, E. coli, Pseudomonas aeruginosa,* and

the *fungus Candida albicans* [24]. The beneficial properties of this plant are attributed to bioactive molecules (secondary metabolites), including phenolic compounds, tannins, alkaloids, glycosides, sterols and triterpenes, saponins, and others it consists of [24,25]. In Ethiopia, Indigenous traditional healers have been utilizing *V. faba* seed to address various health issues such as body swelling, bronchitis, tapeworm infestations, skin boils, and wounds [26–29]. Additionally, dried seeds are chewed to remedy gastric ulcers [30] and gastritis [16,31,26]. However, there is no scientific evidence in this regard. Therefore, the current study aims to verify the claims made by traditional medicine practitioners regarding the anti-ulcer effect of *V. faba* by investigating its effect in mice. Furthermore, the results of this experimental study will provide valuable insights for the scientific community, encouraging further exploration of this potential medicinal plant. This includes conducting advanced research on drug formulations, exploring the molecular mechanisms of plant-derived treatments, and isolating specific anti-ulcer compounds with scientific support.

## Materials and methods

### Equipment, drugs, chemicals, and supplies

In this experiment, the following chemicals, equipment, and reagents were utilized:

**Chemicals, drugs, and reagents.** Diazepam 10 mg/ml injection (Lab tech chemicals, Gland pharm limited, India), Tween 80 (Uni-chem, India,) Ethyl acetate (Blulux laboratories, Abron, India), Ketamine hydrochloride injection USP (Rotex Medica, Germany), methanol absolute (Alpha Chemika, India), n-hexane (Central drug house (P) Ltd, India), ethyl acetate (Central drug house (P) Ltd, India), distilled water (UOG, Medical Laboratory), chloroform (Nice Chemicals Lab, Kerala), sodium hydroxide (Central Drug House, India), concentrated hydrochloric acid, phenolphthalein (Rankem, India), absolute ethanol (Bululux Laboratories, India), omeprazole 30 mg tab, and misoprostol 200 mcg tablet (Jai Pharma Ltd., India), normal saline 0.9% (Sansheng Pharmaceutical PLC, Ethiopia), Wagner's reagent, formalin 10% buffered solution (Yilmana Chemicals, Ethiopia), Glacial acetic acid (Lobe chemi, India),2% Ferrous and Ferric Chloride,10% sodium hydroxide, concentrated sulfuric acid, benzene, diluted Ammonia. All chemicals used were of analytical grade and were purchased from the Ethiopian Pharmaceutical supply services in Gondar, Ethiopia.

**Equipment and materials.** sensitive digital weighing balance (Abron Exports, India), dry oven (Abron Exports, India), PH-meter, centrifuge machine, test tubes, dropper, burets, deep freezer, lyophilizer, vacuum pump, mortar and pestle, sharp sterilized scissors, forceps, surgical threads with curved needles, round bottom flask, beaker, surgical scalpel blade, aluminum foil, face mask, gauze, Buchner funnel, Whatman filter paper (№ 1) (Okhla industry, New Delhi), gloves, and permanent marker.

### Collection and preparation of the plant material for extraction

The *V. faba* seeds were purchased from a local farmer in Kusquam kebele, Gondar town, 720 km from Addis Ababa [32] at a longitude and latitude of 37.4636° E, 12.6056° N with an altitude of 2,133 meters above sea level on March 15/2024. The plant material was identified and authenticated by Mr. Getenet Chekole (Assistant Professor of Botanical Science in the Department of Biology at the College of Natural and Computational Sciences, University of Gondar). Subsequently, specimens were deposited at the University of Gondar herbarium with voucher 215/07/2024. The required amount of *V. faba* seeds was purchased on March 21, 2024. Then seeds were washed immediately with tap water, dried at room temperature, mechanically powdered, and stored in airtight containers [33].

### Experimental animals

Adult Swiss albino mice of either sex, weighing between 25 and 30 g, and aged 8–12 weeks, were employed in this investigation [34]. The animals came from the Department of Pharmacology's animal houses at the University of Gondar, Ethiopia's School of Pharmacy, College of Medicine and Health Sciences. They were kept at room temperature with a

12:12 h light-dark cycle in cages made of polypropylene plastic, bedding made of wood shavings, and fed a typical pellet diet along with water ad libitum [35]. Before the trial started, the mice were given a week to get used to the lab environment [36]. The established protocols for the use of laboratory animals were adhered to in the care of the animals [37].

## Preparation of hydro-methanol seed extract

**A. crude extract.** The extraction was conducted using an adequate amount of *V. faba* seed powder with 80% methanol in a 1:10 (w/v) dry weight to solvent ratio [38]. The cold maceration technique was employed by soaking the samples in the solvent for three consecutive days (72 h) at room temperature, with frequent manual shaking and stirring. The mixture was then filtered using a muslin cloth, followed by a Whatman (№ 1) filter paper. The residue/marc was re-extracted twice with fresh solvent [39]. The fluid extracts were combined and evaporated to dryness at 40°C using a rotary evaporator to remove methanol. The extract was then dried in a lyophilizer to eliminate any remaining methanol and water. Finally, the dried extract was stored at 4°C in the refrigerator until use.

**B. Fractionation.** Sixty grams of the methanol crude extract were subjected to successive fractionation. The solvents used for the fractionation process were hexane, ethyl acetate, and distilled water, according to their increasing order of polarity [39]. The crude extract was suspended in distilled water in a 1:5 w/v ratio and poured into a separatory funnel. Then an equal volume of n-hexane was added, and the mixture was intermittently shaken to extract hexane-soluble components. The procedure was repeated three times using fresh n-hexane solvent. The n-hexane extract was collected in a beaker [40]. Then, ethyl acetate was added to the aqueous residue, and a similar procedure with n-hexane fractionation was carried out. Then the mixture was shaken and separated into another beaker. The extracts were dried in a rotary evaporator at 40°C, and the yields for each extract were calculated. The aqueous residue was placed in a lyophilizer, and the total aqueous fraction was determined. After drying, the solvent fractions were stored in airtight containers in the refrigerator until their antiulcer activity was evaluated.

## Preliminary phytochemical screening

Preliminary qualitative phytochemical screening tests of secondary metabolites were conducted on the crude extract and the solvent fractions using standard screening guidelines and procedures [41–43].

## Acute oral toxicity test

Following the extraction and fractionation procedures, an acute oral toxicity test was conducted on May 2, and follow-up continued till 17/2024. An acute oral toxicity test was conducted according to the limit test recommendations outlined in OECD Guideline No. 425 [44]. On the initial day of the test, one female Swiss albino mouse, which had fasted for 3 hours, received an oral dose of 2000 mg/kg of the plant extract. The mouse was closely monitored for physical or behavioral changes over the subsequent 24 h, with particular attention during the first 4 hours. No signs of toxicity were evident at this dose. After 24 h from the dosing of the first mice, the other four mice were sequentially treated with the same dose. These mice were monitored daily for any indications of adverse effects, such as changes in appetite, hair erection, lacrimation, tremors, convulsions, salivation, diarrhea, mortality, and other potential symptoms for 14 days [44].

## Grouping and dosing of animals

To investigate the anti-ulcer potential of V. faba on two distinct ulcer models—pyloric ligation-induced and 100% ethanol-induced ulcers—120 mice were randomly assigned to six mouse groups between May 20 and June 15, 2024. The test doses were determined by taking 1/20th of the limit test of the extract as the low dose (100 mg/kg), 1/10th of the limit test of the extract as the middle dose (200 mg/kg), and 1/5th of the limit test of the extract as the third dose (400 mg/kg), according to OECD 425 guideline [45]. Negative control groups received just the solvents of the extracts or fractions [11],

and positive control groups received treatment with omeprazole at 30 mg/kg [34] and misoprostol at 5 µg/kg as a reference standard for ulcers induced by pylorus ligation and 100% ethanol, respectively [46].

## Pylorus ligated induced ulcer model

**A. Single dose.** The Shay rat model, modified by previous studies [44,45], was employed for this investigation. Mice were randomly assigned to five groups, each consisting of six animals. Group 1 was assigned as the negative control and administered a single dose of distilled water (DW, 10 ml/kg). Group 2 was treated with 100 mg/kg of the crude extract, while Group 3 received 200 mg/kg. Group 4 was administered 400 mg/kg of the crude extract. Group 5 was assigned as the reference standard and was administered omeprazole at 30 mg/kg.

**B. Repeated dose.** Mice were randomly divided into five groups, each comprising six animals. All groups were pretreated daily for 10 days. Group 1 was pretreated with DW (10 ml/kg) and served as the negative control. Group 2 was treated with 100 mg/kg of the crude extract, while Group 3 received 200 mg/kg. Group 4 was administered 400 mg/kg of the crude extract. Group 5 was assigned as the reference standard and was given omeprazole at 30 mg/kg.

For both single and repeated pretreatments, mice underwent a 24-h fasting period before the experiment, with access to water permitted until the last 4 h. An hour after the final drug treatment, the mice were anesthetized with 50 mg/kg of ketamine HCl combined with 5 mg/kg of diazepam administered intraperitoneally [47], and a small midline incision below the xiphoid process was made to open the abdomen. The pyloric portion of the stomach was carefully lifted and ligated to prevent traction or damage to the blood supply of the gastric mucosa. The stomach was then carefully repositioned, and the abdominal wall was closed with interrupted sutures using Meri silk no. 2 [46]. After four hours of pyloric ligation, the mice were euthanized using 200 mg/kg of ketamine HCl via intraperitoneal injection [48]. The abdomen was opened along the greater curvature, and its content was drained into a test tube. The gastric juice was collected and centrifuged at 2000 rpm for 10 min [47]. The volume of the resulting supernatant was recorded, and this supernatant was used to determine the total acidity and pH. The stomach mucosa of each animal was washed with saline and clean water, labeled, and preserved in sodium phosphate-buffered 10% formalin until examination for lesions using a hand lens (10×) with subsequent scoring [49].

## Ethanol-induced ulcer model

**A. Single dose.** Ulcers were induced by administering absolute ethanol, following the method outlined by Hollander D. et al. [50] and previous studies with some modifications to assess the anti-ulcer effects of repeated and single-dose administrations [45,46]. Mice were randomly allocated to five groups, each consisting of six animals. Group 1 administered a single dose of DW (10 ml/kg) and was assigned as the negative control. Group 2 was pretreated with 100 mg/kg of crude extract, while Group 3 was pretreated with 200 mg/kg. Group 4 was pretreated with 400 mg/kg of the crude extract. Group 5 was assigned as the reference standard and was pretreated with a single dose of misoprostol at 5 µg/kg.

**B. Repeated dose.** Animals were randomly allocated into five groups, each consisting of six animals. All groups were pretreated daily for 10 days. Group 1 administered a single dose of DW (10 ml/kg) and was assigned as the negative control. Group 2 was pretreated with 100 mg/kg of crude extract. Group 3 was pretreated with 200 mg/kg of the crude extract. Group 4 was pretreated with 400 mg/kg of the crude extract. Group 5 was pretreated with a single dose of misoprostol at 5 µg/kg and assigned as the reference standard.

For both single and repeated dose treatments, respective pretreatments were administered orally to mice that fasted for 24 h. After one hour of pretreatments, an ulcer was induced by administering absolute ethanol (99.9% w/v) at 1 ml/200 g body weight for each mouse [46]. After one hour of ethanol administration, animals were euthanized using anesthetics with 200 mg/kg of ketamine via intraperitoneal injection [48]. The stomachs were incised along the greater curvature, and

ulceration was scored using a method similar to that outlined by pylorus ligation [51]. Finally, randomly selected photographs of the opened stomach from each group (6 mice) showing visual differences in the severity of ulceration between the negative control, the graded dose of the crude extract, and the reference standard drug are displayed.

**Solvent fraction antiulcer effect study.** After evaluating the effect of the crude extract in pylorus ligation and ethanol-induced ulcer models, as the crude extract was found to show greater ulcer protection in the ethanol-induced ulcer model, the model was selected to study the effect of the solvent fractions. This selection was based on a previous study [47]. Sixty-six mice were randomly allocated into 11 groups, each comprising six animals. Group 1 was assigned as the negative control, receiving only the vehicle (2% Tween 80). Groups 2, 3, and 4 were pretreated with 100 mg/kg, 200 mg/kg, and 400 mg/kg of the aqueous fraction, respectively. Groups 5, 6, and 7 were pretreated with 100 mg/kg, 200 mg/kg, and 400 mg/kg of the ethyl acetate fraction, respectively. Groups 8, 9, and 10 were pretreated with 100 mg/kg, 200 mg/kg, and 400 mg/kg of the n-hexane fraction, respectively. Group 11 was assigned as the positive control and pretreated with misoprostol at 5 µg/kg. Respective pretreatments were administered orally to mice that fasted for 24 h. One hour after the pretreatments, ulcers were induced by administering 1 ml of absolute ethanol (99.9% w/v) per 200 g of body weight to each mouse. After one hour of ethanol administration, the mice were then euthanized using anesthetics with 200 mg/kg of ketamine via intraperitoneal injection. The stomachs were incised along the greater curvature, and ulceration was scored using a method similar to that outlined by pylorus ligation.

## Parameters for evaluation of antiulcer activity

**Macroscopic evaluation of the stomach.** The stomachs were opened along the greater curvature and rinsed with saline and clean water to remove gastric contents and blood clots. The mucosa of each animal was labeled and placed in 10% formalin buffered with sodium phosphate for examination. The tissues were then assessed using a 10 × magnifier lens to evaluate ulcer formation, and the number of ulcers was counted. Ulcers were scored based on the method described by Kulkarni [52] as normal-colored stomach (0), red coloration (0.5), spot ulcer [1], hemorrhagic streaks (1.5), deep ulcers [2], and perforation [3]. The mean ulcer score for each animal was expressed as the ulcer index. The ulcer index (UI) was measured using the following formula:

First, the US was calculated by dividing the sum of the severity score by the number of ulcers per mouse.

$$UI = UN + US + (UP \times 10^{-1})$$

UI is the ulcer index, UN is the average number of ulcers per animal, US is the average severity score, and UP is the percentage of animals with ulcers (US ≥ 0.5). The percentage inhibition of ulceration was calculated as follows:

$$\% \text{ Protective ratio} = 100 - (UI \text{ pretreated})/(UI \text{ control}) \times 100$$

**Determination of volume of gastric secretion, pH, and total acidity.** Mice were sacrificed, and the contents were emptied into a test tube by opening the abdomen along the larger curvature. After collecting the gastric juice, it was centrifuged for 10 minutes at 2000 rpm. The resultant supernatant's volume was noted. A digital pH meter was then used to determine the pH of the centrifuged stomach output. In a 50 ml conical flask, 0.25 ml of gastric output was diluted with 0.25 ml of distilled water to measure the overall acidity. After adding two drops of phenolphthalein indicator, 0.01 N NaOH was used to titrate the liquid until a persistent pink hue was seen. The volume of 0.01 N NaOH used was recorded. The total acidity was then calculated in milli-equivalents per liter (mEq/L) using the following formula [35]:

$$\text{Total Acidity} = (\text{Volume of NaOH} \times N \times 100 \text{ mEq/L})/0.1$$

where N = normality of NaOH.

## Data analysis

Version 27 of the Statistical Package for the Social Sciences (SPSS) was used for data entry, coding, result cleaning, and analysis. For every parameter, the results were displayed as mean±SEM. Tukey's post hoc multiple comparison tests were used after a one-way analysis of variance (ANOVA) to establish statistical significance. The difference in means between groups with a p-value of less than 0.05 was considered statistically significant.

## Ethical consideration

Ethical clearance was obtained from the Research and Ethics Committee, Department of Pharmacology, University of Gondar, with the reference number SOP4/86/2016. All procedures performed on animals were conducted following the rules and regulations outlined in the Guide for the Care and Use of Laboratory Animals [53].

# Results

## Phytochemical screening test result

The phytochemical screening of the seed of *V. faba* hydromethanol crude extract showed the presence of alkaloids, flavonoids, glycosides, phenolic compounds, plant steroids, saponins, tannins, and terpenoids. Table 1 presents phytochemicals identified in crude extract, aqueous, ethyl acetate, and n-hexane fractions.

## Acute oral toxicity study

An acute oral toxicity test was conducted on female mice per the OECD-425 guidelines (OECD, 2008). The study indicated that the hydromethanolic crude extract of *V. faba* did not result in any deaths at a dosage of 2000 mg/kg within the first 24 h or over the subsequent 14 days. Additionally, physical and behavioral assessments of the mice did not indicate any apparent signs of acute toxicity.

## Effects of hydromethanolic crude extract of *V.faba* seeds on pylorus ligation-induced ulcer

**Single dose study.** The ulcer number ($P<0.05$), ulcer score ($P<0.05$), and ulcer index ($P<0.01$) were all considerably decreased in single-dose tests at 200 mg/kg, offering a 39.83% ulcer index protection in comparison to the negative control. Even more noticeable results were seen with the 400 mg/kg dose of crude extract, which significantly reduced the number of ulcers ($P<0.01$), ulcer score ($P<0.01$), and ulcer index ($P<0.001$), offering 55.80% protection against the ulcer index when compared to the negative control. Omeprazole (30 mg/kg) provided 55.80% ulcer index protection and

**Table 1. Phytochemical screening of *V. faba* seeds crude extract and solvent fractions.**

| Phytochemicals | Test | Crude extract | Aqueous Fraction | Ethyl Acetate Fraction | n-hexane fraction |
|---|---|---|---|---|---|
| Phenols | Ferric chloride test | + | + | + | + |
| Alkaloids | Wagner test | + | + | + | + |
| Glycosides | Keller-Killian test | + | + | + | – |
| Plant Steroids | Salkowski's Test | + | – | – | + |
| Flavonoids | Alkaline reagent test | + | + | + | – |
| Tannins | NaOH test | + | + | + | – |
| Terpenoids | Salkowski test | + | + | + | – |
| Saponins | Forth test | + | + | + | + |
| Anthraquinones | Bontrager's test | – | – | – | – |

Symbol indicates (+), presence of the constituent; (-); absence absence of the constituent.

significantly decreased ulcer number, ulcer score, and ulcer index (P < 0.001) when compared to the negative control. In comparison to the negative control and all extract doses, it also markedly raised the pH of the stomach (P < 0.001) and lowered the volume and overall acidity of the stomach juice (P < 0.001). Table 2 summarizes these results.

Furthermore, Fig 1 shows visual differences in several lesions in opened stomachs between negative control, different test doses of the crude extract, and positive control.

**Repeated dose study.** The crude extract at 100 mg/kg significantly decreased ulcer number (P < 0.05) and score (P < 0.05) in a study with repeated daily administration, whereas 200 mg/kg showed a stronger reduction in ulcer number (P < 0.01) and score (P < 0.01) and ulcer index (P < 0.01). The greatest reductions in ulcer number (P < 0.001), score (P < 0.001), index (P < 0.001), decreased gastric juice volume (P < 0.01), decreased overall acidity (P < 0.05), and higher pH (P < 0.01) were observed at the maximum dose of 400 mg/kg. At 100, 200, and 400 mg/kg, the extract

**Table 2. Effects of Pretreatment with Single-Dose Hydromethanolic Crude Extract of *V. faba* Seeds on Pylorus Ligation-Induced Ulcer Model.**

| Group | UN | US | UI | VoGJ | pH | TA (mEq/L) |
|---|---|---|---|---|---|---|
| NC | 11.67 ± 0.76 | 11.66 ± 0.76 | 22.67 ± 0.30 | 0.62 ± 0.01 | 4.6 ± 00 | 8.67 ± 0.24 |
| VFE100 mg/kg | 9.50 ± 0.82 | 10.17 ± 0.78 | 17.65 ± 0.51 | 0.59 ± 0.01 | 4.2 ± 0.01 | 7.75 ± 0.19 |
| VFE 200 mg/kg | 7.4 ± 0.52a1 | 8.33 ± 0.74a1 | 13.64 ± 0.04a2 | 0.64 ± 0.01 | 4.5 ± 0.01 | 7.77 ± 0.17 |
| VFE 400 mg/kg | 4.50 ± 0.20a2b1 | 5.17 ± 0.37a2b1 | 10.02 ± 0.73a3b2 | 0.58 ± 0.02 | 4.3 ± 0.01 | 7.62 ± 0.10 |
| OM 30 mg/kg | 2.17 ± 0.88a3b1c1 | 1.83 ± 0.54a3b2c1 | 9.41 ± 0.43a3b3c2 | 0.58 ± 0.01a3b3c3d3 | 5.6 ± 0.02 a3b3c3d3 | 6.58 ± 0.14 a3b3c3d3 |

Notes: Values are expressed as mean ± SEM (n = 6); [a]compared to NC, [b]compared to 100 mg/kg VFE, [c]compared to 200 mg/kg VFE, [d]compared to 400 mg/kg VFE; [1]P < 0.05, [2]P < 0.01, [3]P < 0.001. Where: NC, negative control; OM, Omeprazole; TA, total acidity; UI, ulcer index; UN, ulcer number; US; ulcer score; VFE, *Vicia faba* seed crude extract; VoGJ, volume of gastric juice.

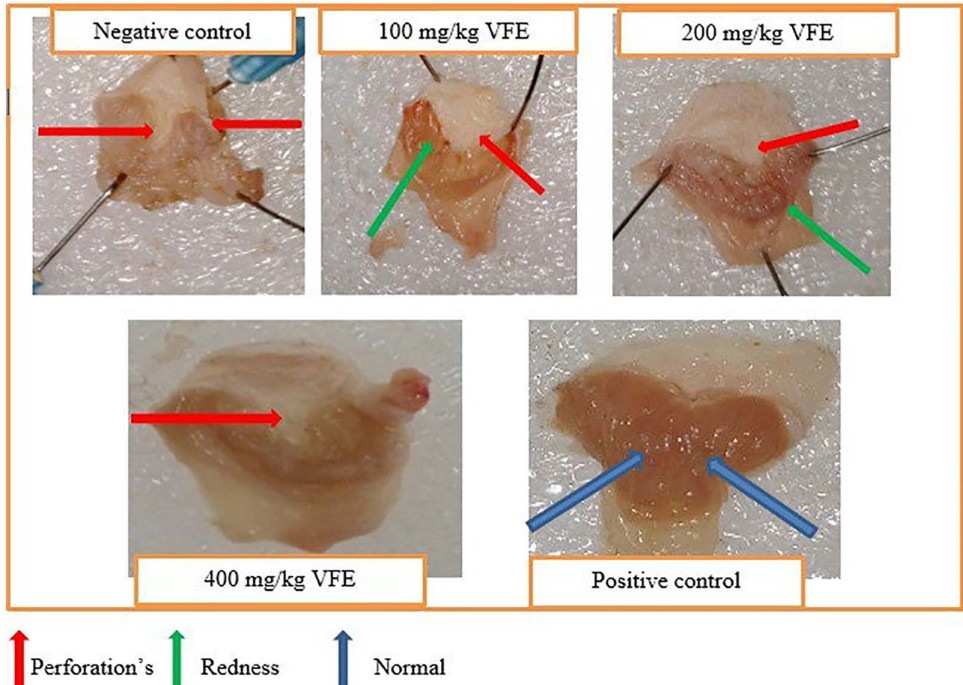

**Fig 1. Effect of pretreatment with single-dose crude extract of *V. faba* on mice's pylorus ligation-induced ulcer model.**

reduced the ulcer index by 33.28%, 44.87%, and 59.17%, respectively. Omeprazole (30 mg/kg) demonstrated the highest anti-ulcer effect with a substantial decrease in ulcer parameters (P < 0.001), decreased gastric juice volume (P < 0.001), decreased total acidity (P < 0.001), and elevated pH (P < 0.001). These effects are summarized in Table 3.

Moreover, Fig 2 displays the crude extract's treating effect, showing fewer ulcers among the group pretreated with the crude extract compared to the negative control.

**Table 3. Effects of pretreatment with repeated daily doses of hydromethanolic crude extract of V. faba seeds on pylorus ligation-induced ulcer model.**

| Group | UN | US | UI | VoGJ | pH | TA (mEq/L) |
|---|---|---|---|---|---|---|
| NC | 12.33±0.56 | 11.17±0.40 | 23.71±0.44 | 0.45±0.01 | 4.4±0.01 | 7.58±0.14 |
| 100 mg/kg | 9.17±0.40a1 | 9.33±0.76a1 | 15.82±0.34a3 | 0.45±0.01 | 4.6±0.01 | 7.47±0.16 |
| 200 mg/kg | 8.33±0.84a2 | 8.50±0.42a2 | 13.07±0.60a3 | 0.43±0.01 | 4.5±0.01 | 7.35±0.14 |
| 400 mg/kg | 4.00±0.13a3b2c1 | 4.50±0.58a3b2 | 9.68±0.04a3 | 0.41±0.01a2b2 | 5.0±0.01a2b1 | 6.93±0.20a1 |
| OM 30 mg/kg | 2.50±0.56a3b3c3 | 2.50±0.72a3b3c3 | 8.48±0.12a3b3c2 | 0.40±0.01a2b3c1 | 6.1±0.01a3b3c3d2 | 6.13±0.13a3b3c3d1 |

Notes: Values are expressed as mean±SEM (n=6). All treatments were given for 10 days for their respective groups. ᵃcompared to NC, ᵇcompared to 100 mg/kg VFE, ᶜcompared to 200 mg/kg VFE, ᵈcompared to 400 mg/kg VFE; ¹P < 0.05, ²P < 0.01, ³P < 0.001. Where: NC, negative control; OM, Omeprazole; TA, total acidity; UI, ulcer index; UN, ulcer number; US; ulcer score; VFE, *Vicia faba* seed crude extract; VoGJ, volume of gastric juice.

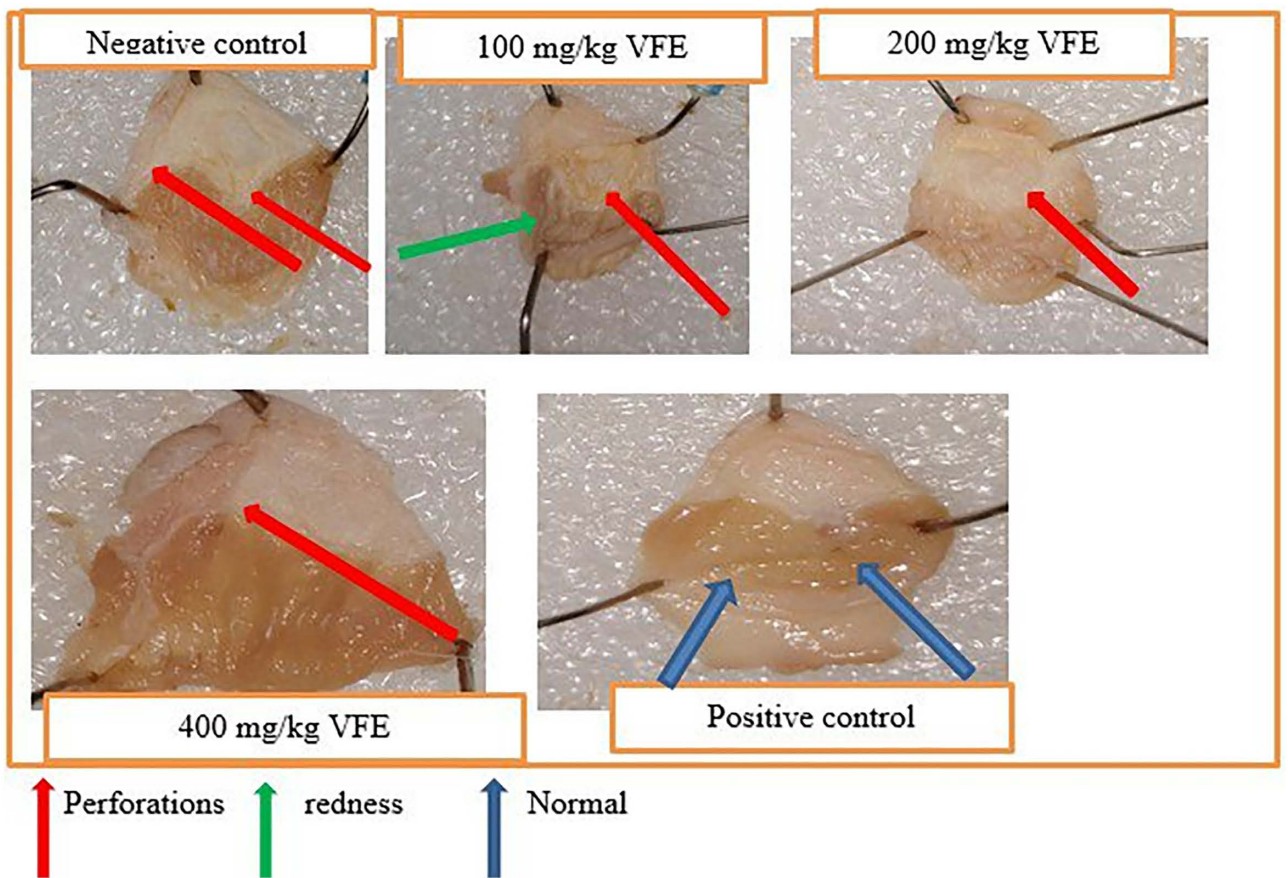

**Fig 2. Effect of pretreatment with repeated daily-dose crude extract of *V. faba* on mice's pylorus ligation-induced ulcer model.**

## Effects of hydromethanolic crude extract of *V.faba* seeds on ethanol-induced ulcer

**Single dose study.** In a single-dose study, the crude extract at 200 mg/kg significantly reduced ulcer number (P<0.01), score (P<0.05), and index (P<0.01). The 400 mg/kg dose showed even greater reductions in ulcer number (P<0.001), score (P<0.01), and index (P<0.01), providing 54.04% protection compared to the negative control. Misoprostol (5 µg/kg) demonstrated the strongest anti-ulcer effects, significantly reducing all parameters at P<0.001 and altering gastric juice metrics. These results are summarized in Table 4.

Fig 3 displays visual differences in the severity of ulceration between negative control, crude extract, and reference standard treated groups.

**Repeated dose study.** At 100 mg/kg, the crude extract dramatically decreased ulcer score and number (p<0.05). The extract decreased ulcer score (p<0.001) and ulcer number (p<0.01) at 200 mg/kg. The most beneficial dose was 400 mg/kg, which changed gastrointestinal parameters (pH, acidity) and dramatically reduced ulcer number (p<0.01), score, and index (p<0.001). At 100, 200, and 400 mg/kg, the crude extract offers 34.94%, 50.91%, and 65.72% ulcer index protection, respectively. Misoprostol also significantly improved all metrics (p<0.001). The detailed results are illustrated in Table 5.

Fig 4 and Fig 5 below show the ulcer index protection of the crude extract in pylorus ligation and absolute ethanol-induced ulcer administered in single and repeated doses; and the severity and number of ulcers in the negative control, group treated with *V.faba* repeated crude dose extract, and groups treated with the standard drug respectively.

## Effects of the repeated dose solvent fractions of *V. faba Seed* crude extract on ethanol-induced ulcer

The aqueous fraction of the crude extract showed significant ulcer inhibition, with 100 mg/kg reducing ulcer metrics (p<0.05), 200 mg/kg further decreasing ulcer number and score (p<0.01), and 400 mg/kg providing the strongest ulcer number and ulcer score reduction (p<0.001). Ulcer index protection was 23.46%, 46.57%, and 62.67% at 100, 200, and 400 mg/kg, respectively. The ethyl acetate fraction also demonstrated efficacy, with 100 mg/kg showing moderate ulcer number and score reduction (p<0.05), and 400 mg/kg significantly reducing ulcer number (p<0.01), score (p<0.05), and index (p<0.001), along with gastric parameter alteration. Ulcer index protection for the ethyl acetate fraction was 21.25%, 40.86%, and 55.21% at 100, 200, and 400 mg/kg, respectively. The n-hexane fraction only reduced scores and index at 400 mg/kg (p<0.05). Misoprostol showed the strongest overall anti-ulcer activity, significantly improving all parameters. The detailed results are presented in Table 6 below.

The comparative ulcer index protection effects of these solvent fractions are illustrated in Fig 6.

## Discussion

The findings of this study showed that the hydromethanolic crude extract of V. faba and its solvent fractions possess promising antiulcer activity in experimental mouse models, supporting its traditional use in Ethiopia. In the acute toxicity

**Table 4. Effects of pretreatment with single-dose hydromethanolic crude extract of V. faba seeds on ethanol-induced ulcer.**

| Group | UN | US | UI | VoGJ | pH | TA (mEq/L) |
|---|---|---|---|---|---|---|
| NC | 11.17±0.48 | 11.67±0.42 | 22.15±0.41 | 0.46±0.01 | 4.03±0.10 | 8.05±0.26 |
| VFE100 mg/kg | 11.00±0.86 | 9.83±0.70 | 16.97±0.97 | 0.44±0.01 | 4.32±0.10 | 7.94±0.08 |
| VFE 200 mg/kg | 8.50±0.05a2 | 8.50±0.81a1 | 14.42±0.75a2 | 0.45±0.01 | 4.34±0.10 | 7.53±0.10 |
| VFE 400 mg/kg | 5.83±0.40a3b2 | 6.00±0.65a3b2 | 10.18±0.06a2b3 | 0.44±0.01 | 4.22±0.10 | 7.50±0.10 |
| MP 5 µg/kg | 2.83±0.95a3b3c2 | 2.50±0.85a3b3c2 | 9.86±0.48a3b3c2 | 0.41±0.01a2b1 | 4.75±0.06a3b1c1d2 | 6.65±0.08 a3b3c2d2 |

Notes: Values are expressed as mean±SEM (n=6); [a]compared to NC, [b]compared to 100 mg/kg VFE, [c]compared to 200 mg/kg VFE, [d]compared to 400 mg/kg VFE; [1]P<0.05, [2]P<0.01, [3]P<0.001. Where: NC, negative control; PC, positive control; TA, total acidity; UI, ulcer index; UN, ulcer number; US; ulcer score; VFE, *Vicia faba* seed crude extract; VoGJ, volume of gastric juice.

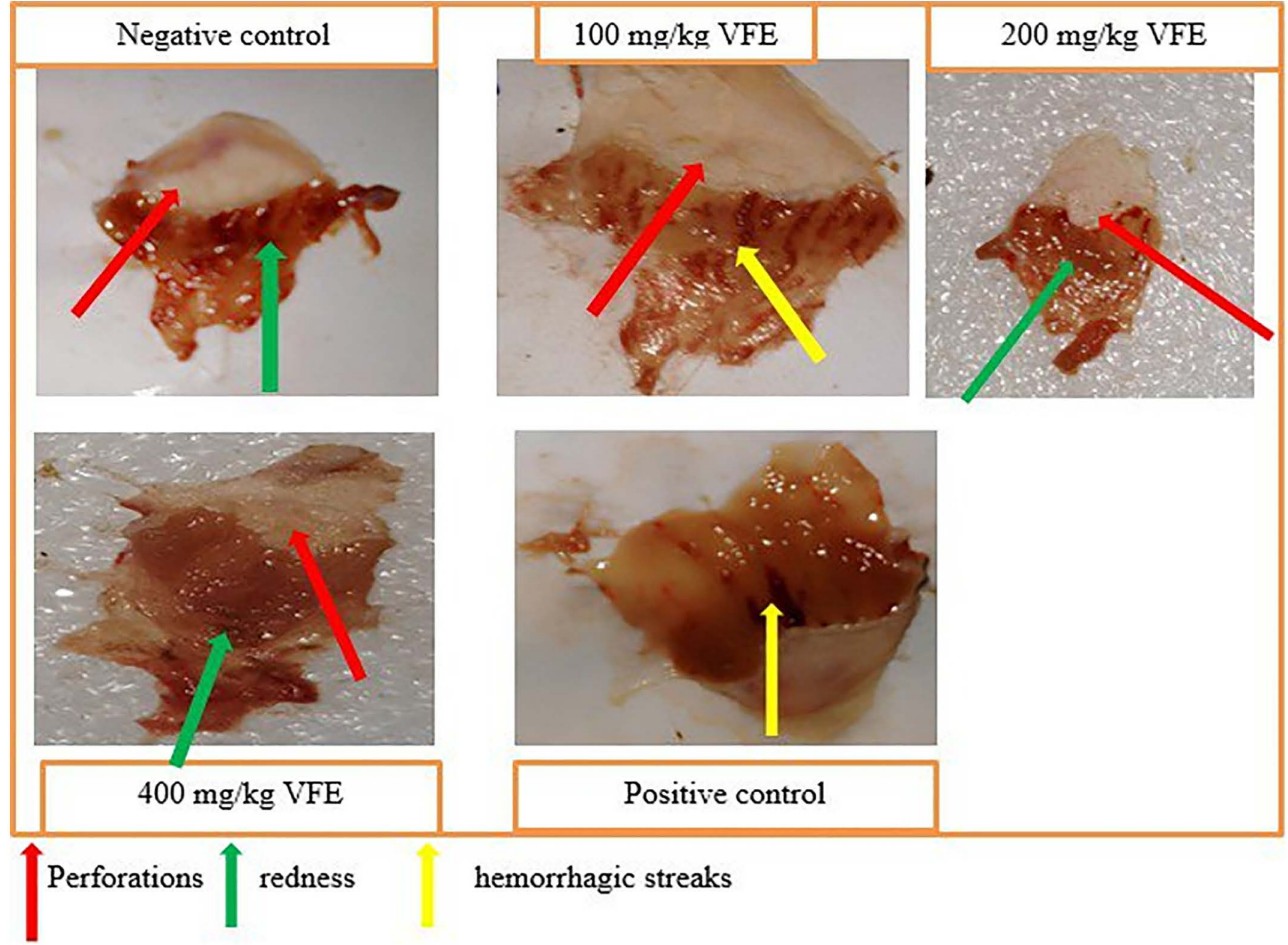

**Fig 3. Effect of pretreatment with single-dose crude extract of *V. faba* on mice's absolute ethanol-induced ulcer model.**

**Table 5. Effects of pretreatment with repeated daily doses of Hydromethanolic Crude Extract of *V. Faba* Seeds on Ethanol-Induced Ulcer.**

| Group | UN | US | UI | VoGJ | PH | TA (mEq/L) |
|---|---|---|---|---|---|---|
| NC | 13.17±0.31 | 14.17±0.48 | 24.24±0.29 | 0.46±0.01 | 4.2±0.01 | 7.58±0.14 |
| VFE100 mg/kg | 9.00±0.86a1 | 8.83±0.75a1 | 15.77±0.62a1 | 0.45±0.01 | 4.5±0.01 | 7.47±0.16 |
| VFE 200 mg/kg | 7.33±0.49a2 | 6.00±0.86a3 | 11.90±0.66a3 | 0.44±0.00 | 4.4±0.01 | 7.35±0.14 |
| VFE 400 mg/kg | 4.67±0.02a3b2 | 4.33±0.92a3b1 | 8.31±0.51a3b2c1 | 0.41±0.01a2b2 | 5.0±0.01a2b1 | 6.93±0.20a1 |
| MP 5 µg/kg | 3.33±0.95a3b3c1 | 2.17±0.70a3b3c1 | 7.82±0.26a3b3c2 | 0.40±0.01a3b3c1 | 6.4±0.01a3b3c3d2 | 6.13±0.13a3b3c3d1 |

Notes: Values are expressed as mean±SEM (n=6); acompared to NC, bcompared to 100 mg/kg VFE, ccompared to 200 mg/kg VFE, dcompared to 400 mg/kg VFE; 1P<0.05, 2P<0.01, 3P<0.001. Where: MP, Misoprostol; NC, negative control; TA, total acidity; UI, ulcer index; UN, ulcer number; US; ulcer score; VFE, *Vicia faba* seed crude extract; VoGJ, volume of gastric juice.

study, administration of the crude extract at a dose of 2000 mg/kg produced no signs of toxicity or mortality during the 14-day observation period. This suggests that V. faba extract is non-toxic, with a median lethal dose ($LD_{50}$) value greater than 2000 mg/kg in mice, indicating it is relatively safe.

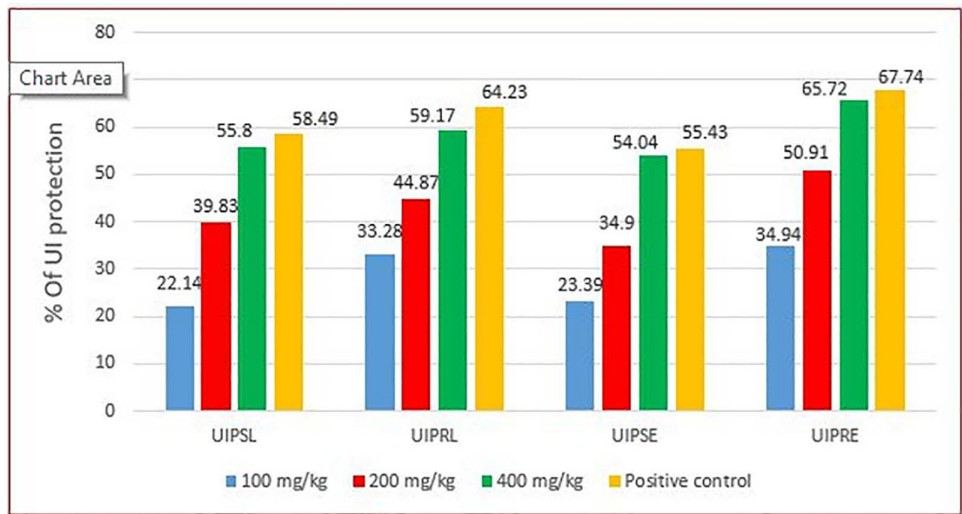

**Fig 4. Percentage of ulcer index protection of pretreatment with a single dose and repeated daily doses of hydromethanolic crude extract of *V. faba* seeds on ethanol and pylorus ligation-induced ulcer models.** UIPSL: ulcer index protection on a single dose of crude extract on pylorus ligation-induced ulcer model; **UIPRL**: ulcer index protection on repeated daily doses of the crude extract on pylorus ligation-induced ulcer model; **UIPSE**: ulcer index protection on a single dose of the crude extract on ethanol-induced ulcer model; **UIPRE**: ulcer index protection on repeated daily doses of the crude extract on ethanol-induced ulcer model.

Most of the phytochemical constituents identified in this study were also identified in other pharmacological research conducted in Egypt [54]. However, this study identified plant steroids and terpenoids, which were not observed in the Egyptian study. The differences in chemical composition could be attributed to variations in soil or climatic conditions, or the specific plant parts used. This study used V. faba seeds, whereas the Egyptian research focused on V. faba bean pods.

In experimental ulcer studies, ulcers can be induced by various methods. In this study, ulcers were induced using pylorus ligation and ethanol induction methods. In both models, repeated dose administration of *V. faba* seed extract produced stronger gastroprotective effects than single doses, likely due to the single-dose administration resulting in a short duration during which the plasma concentration remains within the therapeutic window, potentially failing to achieve the minimum effective plasma concentration. In contrast, repeated doses persistently provide a sufficient concentration of bioactive phytochemicals at the site of action, allowing for sustained therapeutic effects [47]. Higher doses have a more pronounced effect compared to lower doses, likely because they deliver a greater amount of bioactive compounds, which may enhance the modulation of gastric secretions and provide more effective protection against ulcers. The cumulative effect of higher doses could also lead to a more robust and prolonged biological response. Additionally, the highest repeated dose likely produced more significant changes in gastric physiology, such as a reduction in gastric secretion volume, a decrease in total acidity, and an increase in pH. This shows that the antiulcer activity of *V. faba* is mostly attributed to its gastro-protective effect, with minimal anti-secretory effect. This effect is consistent across the study of crude extract in both above mentioned models and aqueous fraction, and ethyl acetate fraction in ethanol-induced ulcers. These findings align with prior studies on other Fabaceae members, such as *Tamarindus indica* and *Calpurnia aurea* seeds (Fabaceae members) in a model of pylorus ligation-induced ulcers [49,55] and Cassia *sieberiana* and *Glycyrrhiza glabra L.* (Fabaceae family members) in a model of ethanol-induced gastric ulcers [34,56], which also demonstrated enhanced protection with higher or repeated dosing.

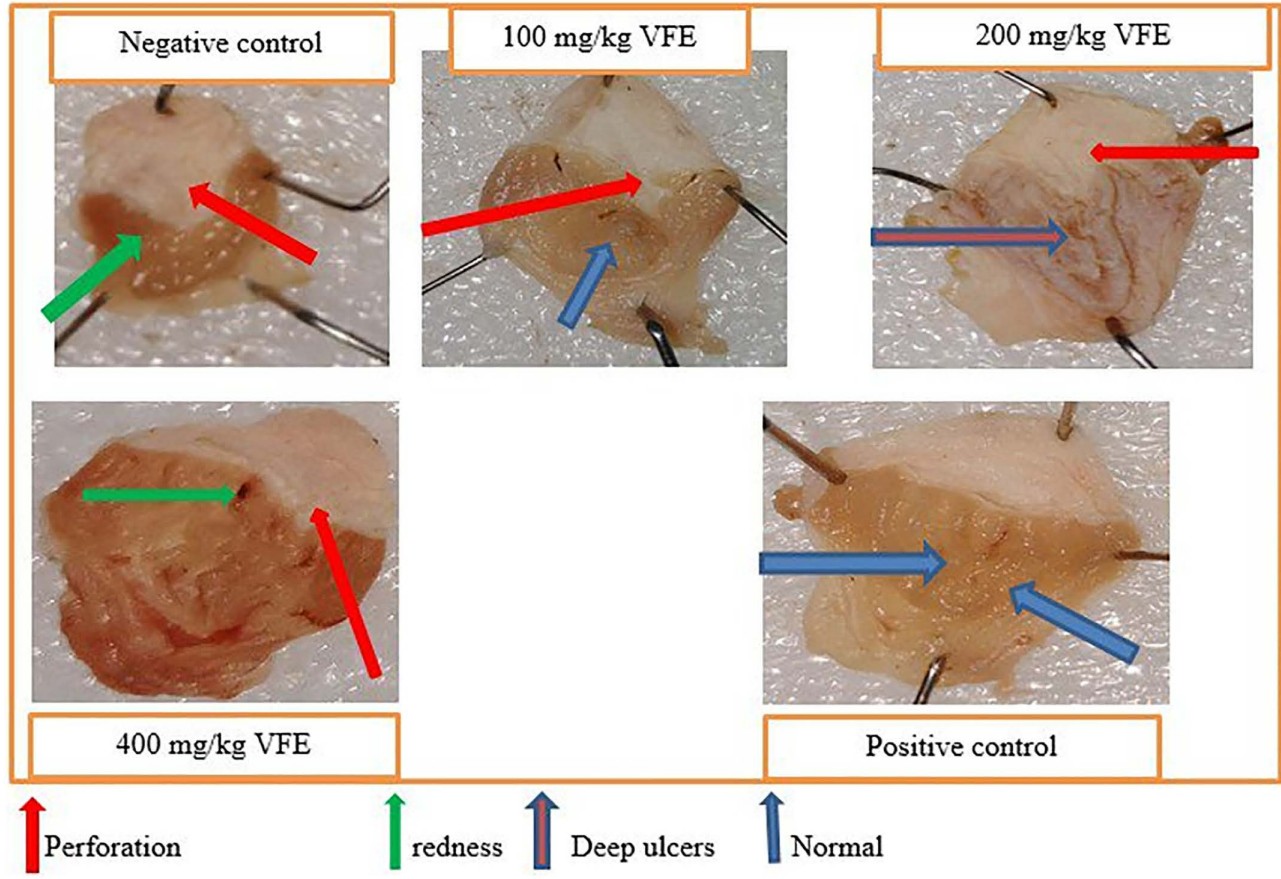

**Fig 5. Effect of pretreatment with daily repeated doses of crude extract of *V. faba* on the absolute ethanol-induced ulcer model.**

In the study of solvent fractions, the lesser antiulcer and anti-secretory effect of the ethyl acetate fraction compared to the aqueous fraction could be due to the lower concentration of the bioactive molecules it consists of [47]. This indicates that most biomolecules were highly polar and more dissolved in the aqueous (polar) solvent. The lack of an anti-secretory effect and the need for higher doses of n-hexane fraction for antiulcer activity also could be explained by the lack of some of the bioactive molecules, including glycosides, terpenoids, flavonoids, and tannins that were identified in the crude extract [49].

As identified by phytochemical assay, the hydromethanolic extract of *V. faba* contains a range of bioactive components, including flavonoids, tannins, phenolic compounds, alkaloids, terpenoids, glycosides, plant steroids, and saponins. Various previous studies have shown that *V.faba* is rich in polyphenolic compounds [57–60]. These polyphenolic compounds are known to have antioxidant activity, possibly by inhibiting different enzymes that are responsible for the generation of various reactive oxygen species. The polyphenolic compounds found in *V.faba* were found to have xanthine oxidase inhibiting activity, an enzyme responsible for the generation of reactive oxygen species [61,62]. Tannins help form a protective barrier at ulcer sites, shielding against toxic substances and proteolytic enzymes [63]. Flavonoids act as free radical scavengers and possess cytoprotective and anti-ulcerogenic properties [10]. Moreover, previous studies have shown that *V. faba* has flavonoids with established antiulcer activity , namely kaempferol and quercetin [64]. Moreover, previous studies have shown that *V. faba* has flavonoids with established antiulcer activity, namely kaempferol and quercetin [65]. Kaempferol

**Table 6. Effects of pre-treatment with repeated daily doses of n-hexane, ethyl acetate, and distilled water fraction of *V. faba* seeds on ethanol-induced ulcer.**

| Group | UN | US | UI | VoGJ | pH | TA (mEq/L) |
|---|---|---|---|---|---|---|
| NC | 12.33±0.92 | 12.00±0.89 | 23.31±0.91 | 0.46±0.10 | 4.55±0.08 | 8.22±0.20 |
| AQ 100 mg/kg | 9.83±0.60a1 | 10.67±0.56a1 | 17.81±0.45a1 | 0.45±0.10 | 4.67±0.08 | 7.92±0.21 |
| AQ 200 mg/kg | 9.17±0.48a2 | 9.33±0.33a2 | 13.98±0.82a2e1h1 | 0.45±0.10 | 4.77±0.08 | 7.58±0.7 |
| AQ 400 mg/kg | 7.67±0.49a3e2h2 | 7.50±0.50a3e3h1 | 10.72±0.24a3b2e1h1i1j1 | 0.40±0.10a1e1h1i1j1 | 4.90±0.04 a1e1h1i1j1 | 7.10±0.13a1e1h1 |
| ET 100 mg/kg | 11.67±0.61 | 12.00±0.73 | 18.38±0.24a2 | 0.46±0.10 | 4.52±0.09 | 7.98±0.13 |
| ET 200 mg/kg | 9.17±0.48a1 | 9.00±0.51 | 13.79±0.54a2 | 0.45±0.10 | 4.67±0.05 | 7.65±0.11 |
| ET 400 mg/kg | 8.33±0.49a2e1h1 | 8.50±0.76a1e1 | 10.44±0.56a3b2e2h2i2 | 0.41±0.10a1e1h1 | 4.80±0.12a1h1 | 7.30±0.11a1e1 |
| NH 100 mg/kg | 11.50±0.76 | 11.00±0.84 | 19.40±0.74 | 0.45±0.0.10 | 4.67±0.05 | 8.07±0.14 |
| NH 200 mg/kg | 9.67±0.70 | 9.50±0.84 | 18.42±0.34 | 0.44±0.10 | 4.72±0.07 | 7.95±0.09 |
| NH 400 mg/kg | 9.72±0.30 | 9.66±0.40a1 | 16.02±0.63a1 | 0.45±0.10 | 4.68±0.09 | 7.63±0.07 |
| MP 5 µg/kg | 2.50±0.76k3 | 2.33±0.76k3 | 8.50±0.07k3 | 0.39±0.10k2 | 6.35±0.10k3 | 6.85±0.10k3 |

Notes: Values are expressed as mean±SEM (n=6); [a]compared to NC, [b]compared to 100 mg/kg AQ, [c]compared to 200 mg/kg AQ, [d]compared to 400 mg/kg AQ; [e]compared to 100 mg/kg ET, [f]compared to 200 mg/kg ET; [g]compared to 400 mg/kg ET; [h]compared to 100 mg/kg NH; [i]compared to 200 mg/kg NH; [j]compared to 400 mg/kg NH; [k]compared to all other fractions and NC [1]$P<0.05$, [2]$P<0.01$, [3]$P<0.001$. Where: AQ, aqueous fraction; ET, ethyl acetate fraction; MP, Misoprostol; NC, negative control; NH, n-hexane fraction; TA, total acidity; UI, ulcer index; UN, ulcer number; US, ulcer score; VFE, *Vicia faba*; VoGJ, volume of gastric juice.

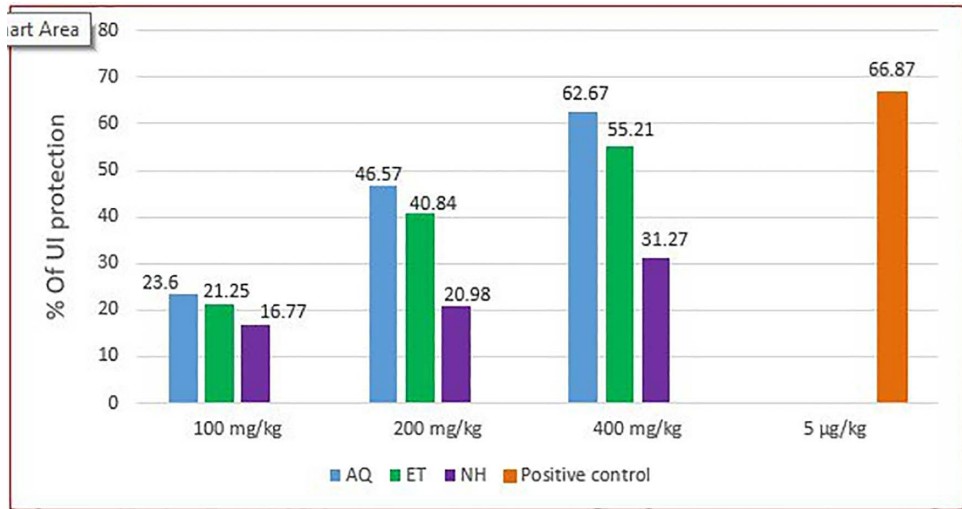

**Fig 6. Percentage of ulcer index inhibition of repeated daily doses of aqueous, ethyl acetate, and n-hexane fractions of *V. faba* Seeds on Ethanol-Induced Ulcer.** AQ, aqueous fraction; ET, ethyl acetate fraction; NH, n-hexane fraction.

reduces myeloperoxidase activity, limiting neutrophil infiltration and minimizing gastric injury, thereby promoting healing and reducing inflammation in the gastric mucosa [66]. The reduction in acid secretion could be attributed to the flavonoids it consists of, which are known to lower histamine release from mast cells by blocking histidine decarboxylase and inhibiting the H+/K+-ATPase enzyme [10,63] and alkaloids, which exhibit an anti-secretory effect by acting as H2-receptor antagonists and through their anticholinergic action [63]. So, the antiulcer activity of *V. faba* crude extract and solvent fractions could be due to its antioxidant, anti-inflammatory activity, and minimal anti-secretory activity. Therefore, *V. faba* could be a promising candidate for treating PUD due to its antioxidant and anti-inflammatory properties, directly targeting the

disease's pathogenesis. So, it is crucial to preserve the genome stability of *V. faba* to maintain its therapeutic potential and genetic integrity. This is especially important given the widespread use of chemicals like pesticides and herbicides that could potentially alter the plant's genetic makeup; thus, using organic and less toxic alternatives and regularly monitoring chemical application is advisable.

### Strengths and limitations of the study

This study provides preliminary evidence for the antiulcer potential of *V. faba* seed extract using pylorus ligation and ethanol-induced ulcer models. Both single and repeated doses of the crude extract and solvent fractions were tested, highlighting the influence of dose and treatment duration. However, this study is not without limitations. The limitations of this study include that while acute toxicity tests suggest safety, long-term effects remain unexamined. Ulcer scoring via the Kulkarni method does not account for ulcer size or area, which can be seen as a limitation. The study also lacked quantitative phytochemical analysis. Furthermore, this study lacks histological (microscopic) analysis, which limits the ability to fully assess tissue-level healing and inflammatory changes, thereby adding greater depth to the interpretation of ulcer healing. Most importantly, as this study was limited to mouse models, the relevance of these results to humans remains uncertain. Advanced preclinical studies, including in higher-order models, and well-designed clinical trials are essential to confirm the therapeutic potential of *V. faba* seeds.

## Conclusion

The present study showed that the hydromethanolic crude extract of *V. faba* and its solvent fractions generally have significant gastroprotective activity in mouse models of gastric ulcer at 400 mg/kg single dose and 200 and 400 mg/kg repeated doses, with repeated dosing proving a more effective treatment. These findings offer preliminary support for the traditional use of *V. faba* seeds against gastric ulcers, but they should be interpreted cautiously given the reliance on animal models. Further studies, including phytochemical characterization, chronic toxicity testing, histological analyses, and well-designed clinical trials, are needed to confirm their safety and efficacy in humans.

## Supporting information

**S1 File  Parameters and their abbreviations used for the evaluation of anti-ulcer activity of hydromethanol crude extract and solvent fractions of *Vicia faba* (Fabaceae) seeds in mice.**
(XLSX)

## Acknowledgments

The authors would like to acknowledge the University of Gondar.

## Author contributions

**Conceptualization:** Demis Getachew, Alemante Tafese Beyna, Mohammedbrhan Abdelwuhab, Assefa Belay Asrie.

**Data curation:** Demis Getachew, Alemante Tafese Beyna, Mohammedbrhan Abdelwuhab, Assefa Belay Asrie.

**Formal analysis:** Demis Getachew, Alemante Tafese Beyna, Mohammedbrhan Abdelwuhab, Assefa Belay Asrie.

**Funding acquisition:** Demis Getachew.

**Investigation:** Demis Getachew, Alemante Tafese Beyna, Mohammedbrhan Abdelwuhab, Assefa Belay Asrie.

**Methodology:** Demis Getachew, Alemante Tafese Beyna, Mohammedbrhan Abdelwuhab, Assefa Belay Asrie.

**Project administration:** Demis Getachew, Alemante Tafese Beyna, Mohammedbrhan Abdelwuhab, Assefa Belay Asrie.

**Resources:** Demis Getachew.

**Software:** Demis Getachew, Alemante Tafese Beyna, Assefa Belay Asrie.

**Supervision:** Mohammedbrhan Abdelwuhab, Assefa Belay Asrie.

**Validation:** Demis Getachew, Alemante Tafese Beyna, Mohammedbrhan Abdelwuhab, Assefa Belay Asrie.

**Visualization:** Demis Getachew, Mohammedbrhan Abdelwuhab, Assefa Belay Asrie.

**Writing – original draft:** Demis Getachew, Alemante Tafese Beyna, Mohammedbrhan Abdelwuhab, Assefa Belay Asrie.

**Writing – review & editing:** Demis Getachew, Alemante Tafese Beyna, Mohammedbrhan Abdelwuhab, Assefa Belay Asrie.

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
