## [Decision Letter · Decision Letter 0]

27 Feb 2025

PONE-D-24-53911Evaluation of Anti-Ulcer Activity of Hydromethanol Crude Extract and Solvent Fractions of Vicia faba (Fabaceae) Seeds in Mice.PLOS ONE

Dear Dr. Getachew,

Thank you for submitting your manuscript to PLOS ONE. After careful consideration, we feel that it has merit but does not fully meet PLOS ONE’s publication criteria as it currently stands. Therefore, we invite you to submit a revised version of the manuscript that addresses the points raised during the review process.

We look forward to receiving your revised manuscript.

Kind regards,

Muhidin Tahir

Academic Editor

PLOS ONE

Journal Requirements:

“Our research is funded by the University of Gondar, Ethiopia”

Reviewers' comments:

Reviewer's Responses to Questions

**Comments to the Author**

1. Is the manuscript technically sound, and do the data support the conclusions?

Reviewer #1: Yes

Reviewer #2: Yes

2. Has the statistical analysis been performed appropriately and rigorously? 

Reviewer #1: No

Reviewer #2: Yes

3. Have the authors made all data underlying the findings in their manuscript fully available?

Reviewer #1: No

Reviewer #2: Yes

4. Is the manuscript presented in an intelligible fashion and written in standard English?

Reviewer #1: No

Reviewer #2: Yes

5. Review Comments to the Author

Reviewer #1: While the study suggests potential for developing new herbal medicines with anti-ulcer effects its ethno pharmacological relevance, however, the results are preliminary and the mechanisms of action were not well explored. I do not support it in this instance. However, I suggest further revisions in the manuscript

î The manuscript has many grammatical and language flaws to be edited with a native speaker.

î I feel concerned with the writing and the English, in which sometimes seems to make no sense

î In preparation of plant seed extract why do you prefer hydromethanol? How is it prepared in the traditional medicine?

î Page 12: Table 1: Qualitative phytochemical analysis of V. faba seeds crude extract and solvent fractions. (I suggest discussing the chemical composition of V. faba after identifying by HPLC?)

î What do you attribute this reduction to? Chemistry composition?

î Stress a notorious cause of ulcer was not represented in this study. Do you come up with a clear reason?

î The authors should describe how doses were chosen for the study

î Due to English language issues results are hard to follow.

î On the whole, practically no discussion was elaborated as mere results were just mentioned again and again. In many part of the discussion, the authors discussed many phytochemicals which they never studied in this work. The preliminary phytochemical study cannot be used as a justification, since, as the name suggests and which it is is 'preliminary', there is need to do fingerprinting of the plant or at least quantification of some of the important phytoconstituents like polyphenols, flavonoids etc.

Thank you

Reviewer #2: 

**I found that the topic was interesting because I am one of individuals suffered with gastric ulcers****!****Comments for authors**

Vicia faba should be italicized throughout the manuscriptLine 115: What plant material was identified by the botanist? B/c you have bought the *Faba bean* seeds from market…..line113?muslin cloth??........line137Line 168: and there were no dead mice or observed signs of toxicity; parts of resultLine 173: plant extract/seed extract???Line 199; After four h. of pyloric ligation?Laboratory study period should be included?Line 411-19: discussion portion irrelevant?? Discuss you finding with previous work rather than introducing methods and objectives?Line 507-518: Conclusion:* V. faba *seeds crude extract and solvent fractions have significant antiulcer activity administered at single and repeated doses …..at what concentrations??**Important question:**Mostly the Ethiopian community utilizes *Faba bean* for consumption in cooked form rather than raw seed but your experiment involves raw powdered *Faba bean *against laboratory mice? Can heat affect the peptic ulcer healing property of *Faba bean*/not??  B/c you observed the uncooked powdered *Faba bean* seeds?

Important points to be considered:

Mostly the Ethiopian community utilizes Faba bean for consumption in cooked form rather than raw seed but your experiment involves raw powdered Faba bean against laboratory mice? Can heat affect the peptic ulcer healing property of Faba bean/not?? B/c you observed the uncooked powdered Faba bean seeds?

6. PLOS authors have the option to publish the peer review history of their article (what does this mean?). If published, this will include your full peer review and any attached files.

Reviewer #1: No

Reviewer #2: No

---

## [Author Response · Author response to Decision Letter 1]

6 Mar 2025

Subject: Submission of Point-by-Point Response for Manuscript ID PONE-D-24-53911

Dear PLOS ONE Editors:

I hope this message finds you well. I am writing to submit the point-by-point response to the reviewers' comments for Manuscript ID PONE-D-24-53911, entitled “Evaluation of Anti-Ulcer Activity of Hydromethanol Crude Extract and Solvent Fractions of Vicia faba (Fabaceae) Seeds in Mice.” Attached to this email, you will find our comprehensive response addressing each of the reviewers' comments. We have carefully considered their feedback and made the necessary revisions to improve the quality and clarity of the manuscript. We appreciate the time and effort the reviewers have dedicated to evaluating our work and providing valuable insights. We believe that the revisions we have made address their concerns effectively. Please do not hesitate to reach out if you require any further information or clarification regarding our response. We look forward to your feedback on the revised manuscript. Thank you for your continued support and guidance throughout the review process.

Sincerely,

Demis Getachew Shawule

University of Gondar,

Gondar, Ethiopia

demisgetachew1@gmail.com

Point-by-point correction and response to the reviewer

General Comments:

Dear Editors and reviewer

Thank you for the constructive comments. We have revised the manuscript according to the journal's guidelines and comments.

Comments to the Author

Comments: 1. Is the manuscript technically sound, and do the data support the conclusions?

Reviewer #1: Yes

Responses: Thank you

Reviewer #2: Yes

Responses: Thank you

Comments: 2. Has the statistical analysis been performed appropriately and rigorously?

Reviewer #1: No

Responses: We believe the statistical analysis was performed rigorously. We conducted an ANOVA (Analysis of Variance) to assess whether there were significant differences between the groups. Following the ANOVA, We performed a Tukey post-hoc test to determine which specific group pairs differ significantly from one another. Both tests are appropriate for this type of analysis, as ANOVA helps identify overall group differences, while Tukey's test ensures that any pairwise differences are evaluated while controlling for the risk of Type I errors.

Reviewer #2: Yes

Responses: Thankyou

Comments: 3. Have the authors made all data underlying the findings in their manuscript fully available?

Reviewer #1: No

Responses: We think we made sufficient data underlying the findings available in the manuscript. Moreover, the full data set was submitted to the journal as Supplementary file1 during manuscript submission.

Reviewer #2: Yes

Responses: Thank you

Comments: 4. Is the manuscript presented in an intelligible fashion and written in standard English?

Reviewer #1: No

Responses: Thank you! We have reviewed the entire document and made several changes to fix grammatical errors. We corrected some grammatical errors using QuillBot's online grammar checker. Additionally, Dr. Yoseph, a senior English language professional (PhD in Teaching English as a Foreign Language (TEFL)) at the University of Gondar, revised the document for language clarity, and further corrections were made based on his recommendation.

Reviewer #2: Yes

Responses: Thank you

5. Review Comments to the Author

Comments: Reviewer #1: While the study suggests potential for developing new herbal medicines with anti-ulcer effects its ethno pharmacological relevance, however, the results are preliminary and the mechanisms of action were not well explored. I do not support it in this instance. However, I suggest further revisions in the manuscript

Responses: Thank you for your valuable comments and suggestions. We revised the manuscript accordingly.

Comments: î the manuscript has many grammatical and language flaws to be edited with a native speaker.

î I feel concerned with the writing and the English, in which sometimes seems to make no sense.

Responses: Thank you for your constructive comments. While we are unable to have a native speaker review the manuscript at this time, as much as possible, we have corrected the grammatical flaws using QuillBot's online grammar checker to improve clarity, coherence, and overall readability. The document was also Dr. Yoseph, a senior English language professional (PhD in Teaching English as a Foreign Language (TEFL)) at the University of Gondar, revised the document for language clarity, and further corrections were made based on his recommendation.

Comments: î In preparation of plant seed extract why do you prefer hydromethanol? How is it prepared in the traditional medicine?

Responses: We preferred hydromethanol due to its expanded polarity index. Hydroalcoholic solvent combinations are often thought to provide good extraction yields. In general, hydroalcoholic co-solvents like 80% methanol appear to have the best solubility properties for initial crude extraction (Reference; Title: Antidiarrheal activities of methanolic crude extract and solvent fractions of the root of Verbascum sinaiticum Benth. (Scrophularaceae) in mice; Authors: Solomon Ayenew Worku, Solomon Asmamaw Tadesse, Mohammedbrhan Abdelwuhab, Assefa Belay Asrie). Furthermore, we found that methanol was used to extract active metabolites of V.faba like polyphenols, and flavonoids (Study conducted in Egypt; Title: Antioxidant, anti-inflammatory, antimicrobial, and anticancer properties of green broad bean pods (Vicia faba L.)). In traditional medicine, hydromethanol was not commonly used to prepare Vicia faba, as its users typically chewed the dry seeds directly to treat stomach ulcers. This explanation is removed in the revised manuscript in response to another reviewer’s comment.

Comments: î Page 12: Table 1: Qualitative phytochemical analysis of V. faba seeds crude extract and solvent fractions. (I suggest discussing the chemical composition of V. faba after identifying by HPLC?)

Responses: Thank you for your feedback. We wholeheartedly acknowledge the necessity and significance of performing HPLC analysis to identify bioactive chemicals with antiulcer activity. However, due to our institution's lack of laboratory facilities that would have allowed us to conduct the research, we could not include data in this respect. We formulated the study's goals from the start, taking into account the available resources and laboratory facilities that can make our plans feasible. Otherwise, we have acknowledged this point in full and do not question the significance of conducting a more sophisticated study that might provide more precise information about the bioactive chemicals that may have an antiulcer effect. As a result, we listed this concern as one of the study's limitations and suggested that more research be done on the plant's chemical constituent and potential molecular mechanisms of action to explain its observed antiulcer effect.

Comments: î What do you attribute this reduction to? Chemistry composition?

Responses: Thank you for your feedback. The reduction to this is attributed to the lack of facilities including HPLC and the chemicals that are used for HPLC function.

Comments: î Stress a notorious cause of ulcer was not represented in this study. Do you come up with a clear reason?

Responses: Thank you for your valuable comment. The omission of stress as a cause of ulcers in this study was intentional. Since this manuscript is part of a thesis, we aimed to keep the focus narrow and concise. As such, we intentionally excluded stress as the third most important stress-inducing factor. However, we have now addressed its role in the pathogenesis of ulcers in the revised manuscript. Please refer to the revised manuscript.

Comments: î The authors should describe how doses were chosen for the study

Responses: Thank you for your deep insight and comment. The test doses were determined by taking 1/20th of the limit test of the extract as the low dose, 1/10th of the limit test of the extract as the middle dose, and 1/5th of the limit test of the extract as the third dose, according to OECD 425 guideline. We explained this in the revised manuscript.

î Due to English language issues results are hard to follow.

î On the whole, practically no discussion was elaborated as mere results were just mentioned again and again. In many part of the discussion, the authors discussed many phytochemicals which they never studied in this work. The preliminary phytochemical study cannot be used as a justification, since, as the name suggests and which it is is 'preliminary', there is need to do fingerprinting of the plant or at least quantification of some of the important phytoconstituents like polyphenols, flavonoids etc.

Thank you

Reviewer #2:

I found that the topic was interesting because I am one of individuals suffered with gastric ulcers!

Comments for authors

Comments: Vicia faba should be italicized throughout the manuscript

Responses: Thank you for your constructive feedback and insightful comments. We have italicized “Vicia faba” throughout the revised manuscript accordingly.

Comments: Line 115: What plant material was identified by the botanist? B/c you have bought the Faba bean seeds from market…..line113?

Responses: Thank you for your constructive comments. First, we brought the seed to a Botanist for identification. Then, the botanist ordered us to cultivate the plant as the whole plant was not available when we were collecting the experimental plant. Accordingly, we brought the seed and the whole plant after cultivating it, and the plant material (the whole plant) was identified and authenticated by Mr. Getenet Chekole, (Assistant Professor of Botanical Science in the Department of Biology at the College of Natural and Computational Sciences, University of Gondar). Subsequently, specimens were deposited at the University of Gondar herbarium with voucher 215/07/2024.

Comments: muslin cloth??........line137

Responses: Thank you for your feedback. Muslin cloth is a loosely woven, cotton fabric that is effective in separating solid particles from liquids. When filtering crude extracts (for example, in the preparation of plant extracts, oils, or juices), muslin cloth can be used to strain out larger solid materials, leaving behind the finer, liquid components. So, we use it in this experiment to separate large solid particles of the macerated bean seed powder from the liquids followed by filter paper filtration.

Comments: Line 168: and there were no dead mice or observed signs of toxicity; parts of result

Responses: Thank you for your deep insight and comment. We have put it in the result section. See the result section of the revised manuscript.

Comments: Line 173: plant extract/seed extract???

Responses: Thank you for your deep insight and comment. It is a writing mistake, we corrected it as “seed extract”.

Comments: Line 199; After four h. of pyloric ligation?

Responses: Thank you for your deep insight and comment. It is a writing mistake. We corrected it as hours

Comments: Laboratory study period should be included?

Responses: Thank you for your feedback. We have included the study period accordingly.

Comments: Line 411-19: discussion portion irrelevant?? Discuss you finding with previous work rather than introducing methods and objectives?

Responses: Thank you for your feedback. We corrected it accordingly. See the revised manuscript.

Comments: Line 507-518: Conclusion: V. faba seeds crude extract and solvent fractions have significant antiulcer activity administered at single and repeated doses …..at what concentrations??

Responses: Thank you for your comments and suggestions. Significant antiulcer activity administered at single and repeated doses of V. faba seeds was observed at 200 and 400 mg/kgs.

Comments: Important question:

Mostly the Ethiopian community utilizes Faba bean for consumption in cooked form rather than raw seed but your experiment involves raw powdered Faba bean against laboratory mice? Can heat affect the peptic ulcer healing property of Faba bean/not?? B/c you observed the uncooked powdered Faba bean seeds?

Important points to be considered:

Mostly the Ethiopian community utilizes Faba bean for consumption in cooked form rather than raw seed but your experiment involves raw powdered Faba bean against laboratory mice? Can heat affect the peptic ulcer healing property of Faba bean/not?? B/c you observed the uncooked powdered Faba bean seeds?

Responses: Thank you for your important comments. Mostly the Ethiopian community utilizes Faba bean for consumption in cooked form rather than raw seed. However, for treatment purposes, they directly chew the dried seed. For example, you can see the ethnobotanical studies that we cited:

Title: An ethnobotanical study of medicinal plants in Wayu Tuka District, East Welega Zone of Oromia Regional State, West Ethiopia.

Authors: Moa Megersa, Zemede Asfaw, Ensermu Kelbessa, Abebe Beyene, and Bizuneh Woldeab

Title: Ethnobotanical study of medicinal plant species in Nensebo District, south-eastern Ethiopia

Authors: Gemedi Abdela, Zerihun Girma and Tesfaye Awas

---

## [Decision Letter · Decision Letter 1]

8 Jul 2025

PONE-D-24-53911R1Evaluation of Anti-Ulcer Activity of Hydromethanol Crude Extract and Solvent Fractions of Vicia faba (Fabaceae) Seeds in Mice.PLOS ONE

Dear Dr. Getachew,

Thank you for submitting your manuscript to PLOS ONE. After careful consideration, we feel that it has merit but does not fully meet PLOS ONE’s publication criteria as it currently stands. Therefore, we invite you to submit a revised version of the manuscript that addresses the points raised during the review process.

We look forward to receiving your revised manuscript.

Kind regards,

Awatif Abid Al-Judaibi, PhD

Academic Editor

PLOS ONE

Journal Requirements:

Additional Editor Comments:

Dear Authors,

Although some reviewers have recommended acceptance of the manuscript, please note that this does not constitute a final editorial decision. The editorial team has carefully reviewed the reviewers’ comments and your responses.

Before proceeding further, we kindly request that you thoroughly address all reviewers’ comments, with special attention to the concerns raised by Reviewer 1 and Reviewer 3.

Reviewer 1 highlighted important issues regarding the preparation of the product and the phytochemical analysis, which were not sufficiently addressed. Reviewer 3 also provided several significant comments that still require revision, despite their general recommendation to accept the manuscript.

Please revise the manuscript accordingly and ensure that each comment is clearly and individually responded to in a point-by-point response letter.

We appreciate your cooperation and look forward to receiving your revised version.

Reviewers' comments:

Reviewer's Responses to Questions

**Comments to the Author**

1. If the authors have adequately addressed your comments raised in a previous round of review and you feel that this manuscript is now acceptable for publication, you may indicate that here to bypass the “Comments to the Author” section, enter your conflict of interest statement in the “Confidential to Editor” section, and submit your "Accept" recommendation.

Reviewer #2: All comments have been addressed

Reviewer #3: All comments have been addressed

Reviewer #4: All comments have been addressed

Reviewer #5: All comments have been addressed

2. Is the manuscript technically sound, and do the data support the conclusions?

Reviewer #2: Yes

Reviewer #3: Yes

Reviewer #4: Yes

Reviewer #5: Yes

3. Has the statistical analysis been performed appropriately and rigorously? 

Reviewer #2: Yes

Reviewer #3: Yes

Reviewer #4: Yes

Reviewer #5: Yes

4. Have the authors made all data underlying the findings in their manuscript fully available?

Reviewer #2: Yes

Reviewer #3: Yes

Reviewer #4: Yes

Reviewer #5: Yes

5. Is the manuscript presented in an intelligible fashion and written in standard English?

Reviewer #2: Yes

Reviewer #3: Yes

Reviewer #4: Yes

Reviewer #5: Yes

6. Review Comments to the Author

Reviewer #2: Comments incorporated! You have selected appropriate area for investigation. Try to to bring the finding of this research to the community.

Reviewer #3: This manuscript presents a meaningful study on the anti-ulcer potential of Vicia faba seed extracts using both ethanol-induced and pylorus ligation models in mice. The experimental design is sound, and the findings support the plant’s traditional use in treating gastric ulcers. The inclusion of both single and repeated dosing regimens is commendable and strengthens the validity of the results.

That said, a few minor points should be considered:

1.While the study shows promising results, a brief discussion of potential mechanisms (e.g., antioxidant or mucosal protection) would enhance the paper.

2.The ulcer severity assessment is based only on visible observation. Adding tissue-level analysis would help support the findings more convincingly. The lack of histological analysis is understandable but should be acknowledged as a limitation.

3.Some language polishing is still needed to improve clarity and flow.

4.A short comment on how the extraction method compares to traditional use would provide helpful context.

Accept with minor edits. The study is valuable and well-conducted. With slight improvements in presentation and clarity, it will make a good contribution to the field.

Reviewer #4: The manuscript has been revised by authors in accordance with reviewers’ comments and suggestions. As per suggestion, the revised manuscript has been supplemented with further relevant information. Various sections particularly experimental as well as discussion are modified for better clarity. The grammatical and linguistic errors have been rectified and presentation of text matter has been improved. Based on the work described, scope of further research is mentioned. Other suggested corrections/modifications have also been aptly incorporated and a detailed clarification has been submitted by author(s) against various queries of reviewers.

Reviewer #5: This is an interesting study conducted by the authors in a scientifically sound manner. The study design is clearly outlined as well as the details of the mice models that were used and the methods used for inducing ulcers. The protocols are clearly outlined and in great detail. The use of tables further adds to the visual quality of the manuscript. However some of the graphs were mentioned in the supplementary data and would be helpful if they are included in the main text itself as that will further and hence the quality of the manuscript. The statistical methods are described in very brief detail, further elaboration would also be helpful. The authors have used appropriate and relevant references to support their study and their discussion. They have also outlined the strength in limitations of the study separately as well.

7. PLOS authors have the option to publish the peer review history of their article (what does this mean?). If published, this will include your full peer review and any attached files.

Reviewer #2: No

Reviewer #3: No

Reviewer #4: No

Reviewer #5: **Yes: **Harleen Kaur Chela

---

## [Author Response · Author response to Decision Letter 2]

17 Jul 2025

Subject: Submission of Point-by-Point Response for Manuscript ID PONE-D-24-53911R1

Dear PLOS ONE Editors:

I hope this message finds you well. I am writing to submit the point-by-point response to the reviewers' comments for Manuscript ID PONE-D-24-53911R1, entitled “Evaluation of Anti-Ulcer Activity of Hydromethanol Crude Extract and Solvent Fractions of Vicia faba (Fabaceae) Seeds in Mice.” Attached to this email, you will find our comprehensive response addressing each of the reviewers' comments. We have carefully considered their feedback and made the necessary revisions to improve the quality and clarity of the manuscript. We appreciate the time and effort the reviewers have dedicated to evaluating our work and providing valuable insights. We believe that the revisions we have made address their concerns effectively. Please do not hesitate to reach out if you require any further information or clarification regarding our response. We look forward to your feedback on the revised manuscript. Thank you for your continued support and guidance throughout the review process.

Sincerely,

Demis Getachew Shawule

University of Gondar,

Gondar, Ethiopia

demisgetachew1@gmail.com

Point-by-point correction and response to the reviewer

General Comments:

Dear Editors and reviewer

Thank you for the constructive comments. We have revised the manuscript according to the journal's guidelines and comments.

Comments to the Author

Additional Editor Comments:

Dear Authors,

Although some reviewers have recommended acceptance of the manuscript, please note that this does not constitute a final editorial decision. The editorial team has carefully reviewed the reviewers’ comments and your responses.

Comments: Before proceeding further, we kindly request that you thoroughly address all reviewers’ comments, with special attention to the concerns raised by Reviewer 1 and Reviewer 3.

Responses: Thank you for your suggestions. We have addressed all the reviewer comments accordingly.

Comments: Reviewer 1 highlighted important issues regarding the preparation of the product and the phytochemical analysis, which were not sufficiently addressed. Reviewer 3 also provided several significant comments that still require revision, despite their general recommendation to accept the manuscript.

Responses: Thank you for your suggestions. We have addressed Reviewer 3's comments accordingly. You can see in the revised manuscript. However, we didn’t find the reviewer 1's comments, and you have also informed us that he is unavailable.

Comments: Please revise the manuscript accordingly and ensure that each comment is clearly and individually responded to in a point-by-point response letter.

Responses: Thank you. We have addressed all the comments accordingly

Review Comments to the Author

Comments: Reviewer #2: Comments incorporated! You have selected appropriate area for investigation. Try to to bring the finding of this research to the community.

Responses: Thank you for your encouraging feedback. We appreciate your positive evaluation of the research area we selected. In response to your suggestion, we have outlined possible strategies for disseminating the findings through academic presentations, open-access publications, and community-focused workshops.

Comments: Reviewer #3: This manuscript presents a meaningful study on the anti-ulcer potential of Vicia faba seed extracts using both ethanol-induced and pylorus ligation models in mice. The experimental design is sound, and the findings support the plant’s traditional use in treating gastric ulcers. The inclusion of both single and repeated dosing regimens is commendable and strengthens the validity of the results.

That said, a few minor points should be considered:

Comments: 1.While the study shows promising results, a brief discussion of potential mechanisms (e.g., antioxidant or mucosal protection) would enhance the paper.

Responses: Thank you for your Suggestions. We have briefly discussed the mechanism of action accordingly. See the revised manuscript.

Comments: 2.The ulcer severity assessment is based only on visible observation. Adding tissue-level analysis would help support the findings more convincingly. The lack of histological analysis is understandable but should be acknowledged as a limitation.

Responses: Thank you for your valuable comments. We fully agree that the addition of histological analysis would have strengthened our findings. Due to resource constraints, we were unable to perform tissue-level histological evaluations in the current study. However, we have now explicitly acknowledged this as a limitation in the Strengths and Limitations section of the revised manuscript.

Comments: 3.Some language polishing is still needed to improve clarity and flow.

Responses: We thank the reviewer for pointing out the need for improved language clarity and flow. In response, we have thoroughly revised the manuscript to enhance its readability and coherence. Grammatical errors and awkward phrasings have been corrected, and sentence structures have been refined for better clarity. We also improved transitions between sections and ensured consistency in terminology throughout the text. We believe these changes have significantly improved the overall presentation and quality of the manuscript.

Comments: 4.A short comment on how the extraction method compares to traditional use would provide helpful context.

In traditional medicine, hydromethanol was not commonly used to prepare Vicia faba, as its users typically chewed the dry seeds directly to treat stomach ulcers (stated in the introduction section of the manuscript). But, we selected based on previous studies that methanol was used to extract active metabolites of V.faba, like polyphenols, and flavonoids (Study conducted in Egypt; Title: Antioxidant, anti-inflammatory, antimicrobial, and anticancer properties of green broad bean pods (Vicia faba L.)). This explanation is removed in the revised manuscript in response to another reviewer’s comment.

Responses:

Accept with minor edits. The study is valuable and well-conducted. With slight improvements in presentation and clarity, it will make a good contribution to the field.

Comments: Reviewer #4: The manuscript has been revised by authors in accordance with reviewers’ comments and suggestions. As per suggestion, the revised manuscript has been supplemented with further relevant information. Various sections particularly experimental as well as discussion are modified for better clarity. The grammatical and linguistic errors have been rectified and presentation of text matter has been improved. Based on the work described, scope of further research is mentioned. Other suggested corrections/modifications have also been aptly incorporated and a detailed clarification has been submitted by author(s) against various queries of reviewers.

Responses: Thank you for the thorough evaluation and constructive feedback.

Comments: Reviewer #5: This is an interesting study conducted by the authors in a scientifically sound manner. The study design is clearly outlined as well as the details of the mice models that were used and the methods used for inducing ulcers. The protocols are clearly outlined and in great detail. The use of tables further adds to the visual quality of the manuscript. However some of the graphs were mentioned in the supplementary data and would be helpful if they are included in the main text itself as that will further and hence the quality of the manuscript. The statistical methods are described in very brief detail, further elaboration would also be helpful. The authors have used appropriate and relevant references to support their study and their discussion. They have also outlined the strength in limitations of the study separately as well.

Responses: Thank you for the encouraging and thoughtful feedback on our manuscript.

---

## [Decision Letter · Decision Letter 2]

19 Aug 2025

PONE-D-24-53911R2Evaluation of Anti-Ulcer Activity of Hydromethanol Crude Extract and Solvent Fractions of Vicia faba (Fabaceae) Seeds in Mice.PLOS ONE

Dear Dr. Getachew,

Thank you for submitting your manuscript to PLOS ONE. After careful consideration, we feel that it has merit but does not fully meet PLOS ONE’s publication criteria as it currently stands. Therefore, we invite you to submit a revised version of the manuscript that addresses the points raised during the review process.

We look forward to receiving your revised manuscript.

Kind regards,

Awatif Abid Al-Judaibi, PhD

Academic Editor

PLOS ONE

Journal Requirements:

Additional Editor Comments:

Dear Authors,

Thank you for your revised submission to PLOS ONE. We acknowledge the improvements you have made in response to the reviewers’ comments. However, after further consideration, we find that additional refinement is still required before the manuscript can move forward.

In particular, while you have revised the Discussion and Conclusion sections, the current phrasing still requires careful adjustment. The conclusions should be presented in a way that accurately reflects the scope of the study, which is limited to animal experiments. Generalizations regarding the efficacy of Vicia faba seeds in humans should be avoided, and the text should instead clearly emphasize the study’s limitations and the need for well-designed clinical investigations to confirm these findings.

We therefore ask that you:

Further refine the Discussion to provide a balanced interpretation of the results without overstatement.

Adjust the Conclusion to ensure consistency with the presented data and to highlight the preliminary nature of the findings.

These refinements will ensure that your manuscript aligns with the journal’s publication standards and will significantly improve its clarity and scientific value. We appreciate your efforts and look forward to receiving a carefully revised version.

Reviewers' comments:

Reviewer's Responses to Questions

**Comments to the Author**

1. If the authors have adequately addressed your comments raised in a previous round of review and you feel that this manuscript is now acceptable for publication, you may indicate that here to bypass the “Comments to the Author” section, enter your conflict of interest statement in the “Confidential to Editor” section, and submit your "Accept" recommendation.

Reviewer #3: All comments have been addressed

2. Is the manuscript technically sound, and do the data support the conclusions?

Reviewer #3: Yes

3. Has the statistical analysis been performed appropriately and rigorously? 

Reviewer #3: Yes

4. Have the authors made all data underlying the findings in their manuscript fully available?

Reviewer #3: Yes

5. Is the manuscript presented in an intelligible fashion and written in standard English?

Reviewer #3: Yes

6. Review Comments to the Author

Reviewer #3: (No Response)

7. PLOS authors have the option to publish the peer review history of their article (what does this mean?). If published, this will include your full peer review and any attached files.

Reviewer #3: No

---

## [Author Response · Author response to Decision Letter 3]

25 Aug 2025

Subject: Submission of Point-by-Point Response for Manuscript ID PONE-D-24-53911R2

Dear PLOS ONE Editors:

I hope this message finds you well. I am writing to submit the point-by-point response to the reviewers' comments for Manuscript ID PONE-D-24-53911R2, entitled “Evaluation of Anti-Ulcer Activity of Hydromethanol Crude Extract and Solvent Fractions of Vicia faba (Fabaceae) Seeds in Mice.” Attached to this email, you will find our comprehensive response addressing each of the reviewers' comments. We have carefully considered their feedback and made the necessary revisions to improve the quality and clarity of the manuscript. We appreciate the time and effort the reviewers have dedicated to evaluating our work and providing valuable insights. We believe that the revisions we have made address their concerns effectively. Please do not hesitate to reach out if you require any further information or clarification regarding our response. We look forward to your feedback on the revised manuscript. Thank you for your continued support and guidance throughout the review process.

Sincerely,

Demis Getachew Shawule

University of Gondar,

Gondar, Ethiopia

demisgetachew1@gmail.com

Point-by-point correction and response to the reviewer

General Comments:

Dear Editors and reviewers

Thank you for the constructive comments. We have revised the manuscript according to the journal's guidelines and comments.

Additional Editor Comments:

Comments: 1. Dear Authors,

Thank you for your revised submission to PLOS ONE. We acknowledge the improvements you have made in response to the reviewers’ comments. However, after further consideration, we find that additional refinement is still required before the manuscript can move forward.

Comments: In particular, while you have revised the Discussion and Conclusion sections, the current phrasing still requires careful adjustment. The conclusions should be presented in a way that accurately reflects the scope of the study, which is limited to animal experiments. Generalizations regarding the efficacy of Vicia faba seeds in humans should be avoided, and the text should instead clearly emphasize the study’s limitations and the need for well-designed clinical investigations to confirm these findings.

Comments: We therefore ask that you:

Further refine the Discussion to provide a balanced interpretation of the results without overstatement.

Responses: Thank you for your valuable feedback. We have carefully revised the Discussion and Conclusion sections to ensure that the statements accurately reflect the scope of our study. We have avoided generalizations regarding the efficacy of Vicia faba seeds in humans and instead emphasized the study’s limitations. The revised text clearly highlights that the findings are restricted to animal experiments and underscores the need for well-designed clinical studies to validate these results.

Comments: Adjust the Conclusion to ensure consistency with the presented data and to highlight the preliminary nature of the findings.

These refinements will ensure that your manuscript aligns with the journal’s publication standards and will significantly improve its clarity and scientific value. We appreciate your efforts and look forward to receiving a carefully revised version.

Responses: We thank the reviewer for this important suggestion. The Conclusion section has been revised to ensure consistency with the presented data and to highlight the preliminary nature of the findings. The revised text now clearly states that the results are based on animal experiments and should be interpreted as preliminary, with further well-designed clinical studies required to validate these observations

---

## [Decision Letter · Decision Letter 3]

24 Sep 2025

Evaluation of Anti-Ulcer Activity of Hydromethanol Crude Extract and Solvent Fractions of Vicia faba (Fabaceae) Seeds in Mice.

PONE-D-24-53911R3

Dear Dr. Demis Getachew,

We’re pleased to inform you that your manuscript has been judged scientifically suitable for publication and will be formally accepted for publication once it meets all outstanding technical requirements.

Kind regards,

Awatif Abid Al-Judaibi, PhD

Academic Editor

PLOS ONE

Reviewer #3:

Reviewers' comments:

Reviewer's Responses to Questions

**Comments to the Author**

1. If the authors have adequately addressed your comments raised in a previous round of review and you feel that this manuscript is now acceptable for publication, you may indicate that here to bypass the “Comments to the Author” section, enter your conflict of interest statement in the “Confidential to Editor” section, and submit your "Accept" recommendation.

Reviewer #3: All comments have been addressed

2. Is the manuscript technically sound, and do the data support the conclusions?

Reviewer #3: Yes

3. Has the statistical analysis been performed appropriately and rigorously? 

Reviewer #3: Yes

4. Have the authors made all data underlying the findings in their manuscript fully available?

Reviewer #3: Yes

5. Is the manuscript presented in an intelligible fashion and written in standard English?

Reviewer #3: Yes

6. Review Comments to the Author

Reviewer #3: (No Response)

7. PLOS authors have the option to publish the peer review history of their article (what does this mean?). If published, this will include your full peer review and any attached files.

Reviewer #3: No

---

## [Editor Report · Acceptance letter]

PONE-D-24-53911R3

PLOS ONE

Dear Dr. Getachew,

I'm pleased to inform you that your manuscript has been deemed suitable for publication in PLOS ONE. Congratulations! Your manuscript is now being handed over to our production team.

Kind regards,

on behalf of

Professor Awatif Abid Al-Judaibi

Academic Editor

PLOS ONE